# Knowledge and Practices on Household Disposal of Unused Antimicrobials in Ho Municipality, Ghana

**DOI:** 10.3390/ijerph22101519

**Published:** 2025-10-03

**Authors:** Thelma Alalbila Aku, Jonathan Jato, Lawrencia Dogbeda Atsu, David Oteng, Inemesit Okon Ben, Samuel Owusu Somuah, Hayford Odoi, Emmanuel Orman, Cornelius Dodoo, Yogini Jani, Araba Ata Hutton-Nyameaye

**Affiliations:** 1Department of Pharmacy Practice, School of Pharmacy, University of Health and Allied Sciences, Ho PMB 31, Ghana; talalbila@uhas.edu.gh (T.A.A.); lawrenciaatsu@gmail.com (L.D.A.); davidoteng297@gmail.com (D.O.); sosomuah@uhas.edu.gh (S.O.S.); 2Department of Pharmacognosy and Herbal Medicine, School of Pharmacy, University of Health and Allied Sciences, Ho PMB 31, Ghana; jjato@uhas.edu.gh; 3Department of Pharmacology and Toxicology, School of Pharmacy, University of Health and Allied Sciences, Ho PMB 31, Ghana; ioben@uhas.edu.gh; 4Department of Pharmaceutical Microbiology, School of Pharmacy, University of Health and Allied Sciences, Ho PMB 31, Ghana; hodoi@uhas.edu.gh (H.O.); cdodoo@uhas.edu.gh (C.D.); 5Department of Pharmaceutical Chemistry, School of Pharmacy, University of Health and Allied Sciences, Ho PMB 31, Ghana; eorman@uhas.edu.gh; 6Centre for Medicines Optimisation Research and Education, University College London Hospitals NHS Foundation Trust, London, NW1 2BU, UK; yogini.jani@nhs.net

**Keywords:** medicine disposal, unused or expired medicines, antimicrobials

## Abstract

Unsafe disposal of unused and expired antimicrobial drugs increases their presence in the environment, thereby contributing to the emergence and spread of antimicrobial resistance. This study addressed the lack of sufficient data on unused and expired antimicrobial disposal practices among peri-urban residents in Ghana. This knowledge–attitude–practice (KAP)-based study offers context-specific insights to inform public health education and antimicrobial disposal policy interventions. A cross-sectional study was conducted among 310 residents in the Ho municipality using a well-structured questionnaire. Data was collected on the knowledge, attitudes, and practices of households on how they dispose of unused and leftover antimicrobials. Origin Pro 2022 software was used to analyze the data. Many respondents were males (*n* = 175, 56.5%) and aged between 18 and 30 years (*n* = 196, 63.2%). About 87.1% (*n* = 270) of the respondents agreed that improper disposal of unused antimicrobials could negatively affect the environment. Most of the respondents (71.9%, *n* = 223) had not received counseling on recommended antimicrobial disposal; 75.5% (*n* = 234) of respondents were not aware of institutions collecting unused or expired medicines; and 73.5% (*n* = 228) had never participated in a medicine-return program. Discarding antimicrobials into household trash bins was the most common way of disposal among respondents. The preferred sites to return unused/leftover antimicrobials were community pharmacies and hospitals. Although respondents showed some knowledge and positive attitudes toward safe antimicrobial disposal, further education is needed. Furthermore, most respondents disposed of antimicrobials in household trash, highlighting the need for take-back programs and community pharmacy-based collection. Incorporating disposal guidance into medication counseling and patient information leaflets can enhance awareness and promote appropriate practices.

## 1. Introduction

Unsafe disposal practices such as flushing medications down toilets, discarding them in household trash, or improper incineration of unused and expired medicines, especially antimicrobials, present significant public health and environmental risks because these practices enhance the rate at which pharmaceutical residues seep into landfills, water bodies, and drainage systems, and this leads to widespread contamination and diverse toxicological effects that endanger human health, terrestrial habitats, and aquatic ecosystems [1,2,3,4,5,6]. In addition, the persistence of antimicrobial residues in the environment exerts selective pressure that promotes the development and spread of antimicrobial resistance, intensifying an already critical global health challenge [7,8].

The World Health Organization has emphasized the urgency of addressing antimicrobial resistance (AMR) through its Global Action Plan, which adopts a “One Health” approach to optimize antimicrobial use, strengthen stewardship programs, and promote education across human, animal, and environmental health sectors [9]. However, a critical yet often neglected aspect of AMR mitigation is the safe disposal of antimicrobials to prevent their accumulation in the environment and subsequent resistance development [10].

The improper handling of unused medications often arises from factors such as over-prescription, non-adherence to prescribed regimens, and a lack of awareness about safe disposal practices. Patients may retain unused medicines for various reasons, including symptom resolution, dosage changes, side effects, or expiration [11]. These medications are frequently disposed of improperly, leading to environmental contamination and the emergence of ecopharmacovigilance, which is a field focused on detecting, assessing, and preventing adverse effects related to pharmaceuticals in the environment [5,12].

In the United States, medicine disposal guidance varies by regulatory agency: the Food and Drug Administration (FDA) recommended flushing certain unused, unwanted, or expired medicines to prevent misuse, while the Environmental Protection Agency (EPA) advised mixing them with undesirable household materials before discarding in trash. Both approaches were contested due to risks of environmental pollution [13]. As a safer alternative, medicine take-back programs have been implemented in countries such as Australia, the United States, and Egypt to reduce household stockpiling and prevent improper disposal [14]. Also in developed settings, structured initiatives like New Zealand’s Dispose of Unwanted Medicine Properly (DUMP) project and Canada’s Environmental Prescription Disposal Program (ENVIRx) promote environmentally sound pharmaceutical waste management [1]. Conversely, many low- and middle-income countries do not have established policies or standardized procedures for handling expired, unused, or unwanted medicines. These gaps are often exacerbated by resource limitations, insufficient regulatory oversight, and weak enforcement of legislative measures [1].

In Ghana, particularly in peri-urban areas like Ho Municipality, the absence of clear policies and guidelines on pharmaceutical waste management exacerbates the problem. Antimicrobials, given their widespread use and environmental persistence, pose unique challenges. While some studies have investigated drug disposal practices in urban settings, limited research exists in peri-urban and rural contexts where waste management systems are often inadequate [7,15]. The lack of comprehensive data on the disposal of unused and expired medicines in Ghana highlights an urgent need for research to inform policy development and community-level interventions. Establishing baseline knowledge, awareness, and disposal practices among residents and healthcare providers is critical to addressing irrational disposal practices. Such data will align with global efforts to combat AMR, thereby contributing to environmental protection and improving public health outcomes.

This study aimed to assess the knowledge, awareness, and practices of households in Ho Municipality regarding the disposal of unused and expired antimicrobials and to provide evidence-based recommendations for promoting safe disposal practices.

## 2. Materials and Methods

### 2.1. Study Design

This was a cross-sectional survey conducted with face-to-face interviews. Data were gathered using a validated, well-structured, and pretested questionnaire to assess the participants’ knowledge, attitudes, practices, and perceptions regarding disposing of unused and leftover antimicrobials in the Ho municipality.

### 2.2. Study Area

This study was carried out in the Ho Municipality in the Volta region of Ghana. The Volta region is one of the sixteen regions of Ghana and is situated in the eastern part of the country. This region shares borders with the Oti region to the north, the Gulf of Guinea to the south, the Volta Lake to the west, and the Republic of Togo to the east. Ho is the regional capital and one of the twenty-five (25) administrative districts in the region. It covers a total land area of 2361 square kilometers, representing 11.5% of the region’s total land area. In 2021, the population in the Ho municipality was 180,420, which accounted for 10.9% of the region’s total population [16].

### 2.3. Study Population and Sample Size

Participants included in this study had been residents in the municipality for at least six months, had used antimicrobial agents in the past six months, and were eighteen years and above. People who were unable or refused to participate in the interview were excluded.

Using Slovin’s formula for sample size determination, an estimated minimum sample size of 399 was obtained from a population of 180,420 [16] residents based on a 95% confidence interval and a margin of error of 5%. At the end of data collection, a total of 310 residents participated, representing a response rate of approximately 78%. The discrepancy from the calculated sample size was primarily due to some residents declining to take part in this study.

### 2.4. Data Collection Process

After a thorough review of the literature, the questionnaire was adapted from three studies [15,17,18] to conform to the precise goals of this study. Questionnaires were given to the residents willing to participate in this study and who met the inclusion criteria. The purpose of this research was verbally explained to the study participants before the questionnaire was administered. The researchers went into the various communities within the municipality to engage the residents. The households were selected through convenience sampling.

The questionnaire was divided into four sections, namely the respondents’ socio-demographics, knowledge of unused medicines, disposal practices of unused antimicrobials at home, and attitude and perception of unused or leftover medicine disposal. The questionnaire had open- and closed-ended questions. The questionnaire was translated into the Ewe and Twi languages for non-English-speaking participants. The researcher read the questions and explained them to the illiterate participants, and their answers were accurately recorded on the surveys.

### 2.5. Quality Control

To ensure face validity and reliability of the questionnaire, a pilot study was conducted with twenty (20) participants, and necessary modifications were made accordingly. To ensure accuracy, the questionnaire was first translated into local languages and then back-translated into English by independent bilingual experts. Three research assistants were trained to collect data from the participants.

### 2.6. Data Handling and Analysis

The data entry template did not include participants’ names; instead, codes or numbers were used to identify them. The data generated was imported into Microsoft Excel spreadsheet software (Microsoft Corporation, Redmond, WA, USA; Version 2508 Build 19127.20240). All electronic data were securely stored on password-protected devices, with access restricted solely to members of the research team to maintain confidentiality. Backup copies of the electronic data were uploaded to a secure cloud storage repository. Data analysis was carried out using OriginPro Pro 2022 software (OriginLab Corporation, Northampton, MA, USA). The data were described using descriptive statistics, including percentages, cross-tabulation analysis, and logistic regression analysis.

## 3. Results

### 3.1. Demography of Respondents

A total of 310 respondents were surveyed, comprising 175 (56.5%) males and 135 (43.5%) females. Most of the respondents were aged between 18 and 30 years (*n* = 196/310, 63.2%), followed by those between 31 and 40 years (*n* = 72/310, 23.2%). Cumulatively, this shows that a more youthful population of respondents was interviewed. The respondents were highly educated, as a significant proportion of them had at least received tertiary education (*n* = 199/310, 64.2%, *χ^2^* = 404.871, *p* < 0.0001), with some having pursued postgraduate education (*n* = 30/310, 9.7%). Others were either employed in the public sector (*n* = 95/310, 30.6%) or the private sector (*n* = 74/310, 23.9%). Most of the respondents with postgraduate education (*n* = 25/30, 83.3%) were working in the public sector. Relatively fewer proportions of them were either self-employed (*n* = 16/310, 5.2%) or not employed (*n* = 26/310, 8.4%). Table 1 below summarizes the key demographic factors of the respondents.

### 3.2. Knowledge and Awareness of Unused Antimicrobials

Table 2 depicts the respondents’ knowledge and awareness concerning the repercussions of unused antimicrobials on drug resistance, environmental sustainability, and health hazards and their associated demographic factors. Generally, respondents demonstrated a good level of awareness across the assessed knowledge domains. The level of agreement ranged between 83.5% and 91.3%, which was in response to the statements that antimicrobial residues in water bodies can affect aquatic life (*n* = 283/310) and that children are the most vulnerable to risks associated with unused antimicrobials (*n* = 283/310).

Further analysis showed some significant demographic predictors. For instance, female respondents [AOR = 1.53 (1.01–2.32), *p* = 0.046], higher educational status of respondent [AOR = 1.22 (1.02–1.45), *p* = 0.027], and respondents’ employment status [AOR = 0.51 (0.26–0.99), *p* = 0.049] demonstrated greater knowledge on the statement that improper disposal of unused antimicrobials can affect health. The female participants [AOR = 1.62 (1.07–2.45), *p* = 0.023] and respondents with higher educational status [AOR = 1.20 (1.00–1.43), *p* = 0.049] were more likely to also agree to the vulnerability of children to risks associated with unused antimicrobials. Privately employed [AOR = 0.54 (0.30–0.96), *p* = 0.034] and self-employed [AOR = 0.52 (0.27–0.99), *p* = 0.049] respondents were less likely to agree that unused or leftover antimicrobials presented potential risks at home (Table 2). 

Providing proper guidance to the consumer (*n* = 283/310, 91.3%) and prescribing adequate quantities of medicines for patient compliance (*n* = 218/310, 70.3%) were the key strategies that most respondents agreed to, minimizing the hazardous effects of unused antimicrobials. While relatively few respondents held that reducing the number of prescribed antimicrobials by the doctor could help address the effects (*n* = 138/310, 44.5%), a minority held that donating or sharing medicines could contribute to minimizing domestic pharmaceutical waste (*n* = 27/310, 8.7%) (Table 3).

Higher educational status was significantly associated with increased support for prescribers to provide adequate quantities of medicines for patient compliance [AOR = 1.29, 95% CI: 1.03–1.62, *p* = 0.027] and advocacy for reduced antimicrobial prescriptions [AOR = 1.47, 95% CI: 1.16–1.84, *p* = 0.001]. In contrast, individuals with higher educational status were less likely to agree with medication donation or sharing [AOR = 0.75, 95% CI: 0.57–0.99, *p* = 0.045]. The privately employed [AOR = 0.41 (0.18–0.88), *p* = 0.022] and self-employed respondents [AOR = 0.32 (0.14–0.78), *p* = 0.012], as well as students [AOR = 0.46 (0.23–0.92), *p* = 0.026], were all less likely to believe that reducing the number of prescribed antimicrobials can control the hazardous effects of unused antimicrobials.

### 3.3. Respondents’ Practices About Antimicrobial Use and Disposal

At the time of this study, a significant proportion of the respondents were not using any antimicrobial (*n* = 286/310, 92.3%; *χ^2^* = 485.6, *p* < 0.0001) (Figure 1). A few of them, however, indicated that they were either on 1–3 antimicrobials (*n* = 20/310, 6.5%) or more than 3 antimicrobials (*n* = 4/310, 1.3%). On the frequency of antimicrobial use, it was also worth noting that a significant majority rarely used them (*n* = 173/310, 55.8%, *χ^2^* = 191.806, *p* < 0.0001). Although those who claimed to use them often (*n* = 37/310, 11.9%) and very frequently (*n* = 14/310, 4.5%) were in the minority. Acute conditions dominated the reasons for the use of the antimicrobials (*n* = 240/310, 77.4%), especially among frequent users (*n* = 8/14, 57.1%). It was also notable that most of the people used antimicrobials rarely or sometimes for acute conditions (*n* = 201/240, 83.8%). Figure 1 summarizes the results of the disposal practices of the respondents.

The primary sources of antimicrobials for the respondents were hospitals (*n* = 189/310, 45.3%) and community pharmacies (*n* = 177/310, 42.4%) (Figure 2). Informal sources included relatives and friends (*n* = 8/310, 1.9%) and medicine peddlers (*n* = 3/310, 0.7%). While most respondents purchased antimicrobials with a prescription (*n* = 214/310, 56.9%), a notable proportion of them had access without a prescription from OTCMSs (*n* = 146/310, 38.8%).

Many of the respondents demonstrated good consumer awareness by routinely checking expiry dates before medicine purchase (*n* = 225/310, 72.6%) (Figure 3). While this is commendable, it is also worth noting that approximately one in four of the respondents (*n* = 85/310, 27.4%) either neglected this practice or only did it occasionally.

Most of the respondents also declared that they had, at some point, possessed some leftover antimicrobials at home (*n* = 230/310, 74.2%), and this was largely due to the discontinuation of treatment regimens (*n* = 189/230, 82.2%; *χ^2^* = 180.11, *p* < 0.0001). The notable reasons assigned to the discontinuation were that there was improvement or resolution of conditions (*n* = 152/206, 73.7%; Figure 3). Also, some respondents discontinued as a result of side effects (*n* = 38/206, 18.4%; Figure 3).

Table 4 offers insights into the disposal practices of antimicrobial drugs among respondents and the demographic factors associated with these practices. The data reported showed that although most of the respondents never shared their medicines (*n* = 210/310, 67.7%), a significant proportion of them engaged in some level of sharing (*n* = 100/310, 32.3%). Respondents with higher educational status were less likely to share their leftover medications with others [AOR = 0.62 (0.40–0.98), *p* = 0.040]. Another concern was that, although most respondents had disposed of antimicrobials in the past (*n* = 234/310, 75.5%), they had not received advice from healthcare providers as to the best disposal practices (*n* = 233/310, 75.2%). Once again, the likelihood to receive disposal advice was strongly associated with educational status [AOR = 1.35 (1.06–1.70), *p* = 0.014].

Additionally, there were observed associations of the demographic factors with the disposal practices with respect to the dosage forms. For tablets/capsules, the most common practice recorded was disposal in household trash (*n* = 202/310, 65.2%). This practice was more likely to be associated with respondents who were self-employed [AOR = 1.89 (1.05–3.41), *p* = 0.034] and less likely to be practiced by respondents with higher education [AOR = 0.82 (0.68–0.99), *p* = 0.038]. The next common disposal practice recorded for tablets/capsules was burning (*n* = 55/310, 17.7%), and this was also more likely to be associated with respondents of older ages [AOR = 1.32 (1.04–1.68), *p* = 0.023] and again less likely to be practiced by respondents with higher education [AOR = 0.73 (0.58–0.92), *p* = 0.008]. In the case of the syrups/suspensions, the common disposal practice observed was also in household trash (*n* = 109/310, 35.2%), and this practice was strongly associated with female respondents [AOR = 1.62 (1.02–2.57), *p* = 0.041]. The higher the educational level of the respondents, the more likely they were to return them to health facilities for safe disposal [AOR = 1.85 (1.12–3.06), *p* = 0.016]. This commendable practice was also observed in the case of ointments/creams [AOR = 2.14 (1.18–3.88), *p* = 0.012] and suppositories/vaginal pessaries [AOR = 2.37 (1.29–4.35), *p* = 0.005].

### 3.4. Attitude and Perception of Unused Antimicrobials

The observations made under this section (Table 5) of the study largely reflect positive attitudes towards the safe use and disposal of antimicrobials, even though some challenges were also observed. Most of them agreed with the idea that purchasing antimicrobials without a prescription can add to the number of unused medicines at home (*n* = 273/310, 88.1%). Pharmacists (*n* = 223/310, 25%), pharmaceutical manufacturers (*n* = 219/310, 24.6%), governmental agencies (*n* = 162/310, 18.2%), and other health professionals (*n* = 183/310, 20.5%) were identified by the respondents as the key responsible persons and institutions for creating awareness for the proper disposal of unused antimicrobials. The institutions responsible for collecting unused medicines from the public to safeguard their health were not known to most of them (*n* = 234/310, 75.5%). The minority (*n* = 76), however, knew of institutions like the Food and Drugs Authority (FDA) (*n* = 51/76, 67.1%), pharmacies (*n* = 43/76, 56.6%), and hospitals (*n* = 38/76, 50.0%), among others. In this regard, they also had either not participated (*n* = 228/310, 73.5%) or were not aware (*n* = 57/310, 18.4%) of the existing medicine-return program. Despite the lack of awareness, most of them were willing to participate in them (*n* = 284/310, 91.6%), and some even reasoned that due to their perceived benefits, they should be made mandatory (*n* = 199/310). To participate in future medicine-return programs, most of them preferred to return their unused medicines to the community pharmacies (*n* = 160/310, 34.3%) or the hospitals or Community Health Planning and Services (CHPS) facilities (*n* = 154/310, 49.7%), and others preferred them being picked up from their homes (*n* = 105/310, 33.9%).

## 4. Discussion

This study found high public knowledge and awareness of AMR, environmental risks, and child safety, consistent with findings from an Indonesian community where, despite generally good knowledge and practices regarding antibiotics, gaps remained in appropriate disposal. This gap may be due to limited awareness of proper disposal methods or the absence of accessible disposal systems. These findings highlight the need for targeted interventions to strengthen safe disposal practices and address residual gaps in antibiotic stewardship [19].

Although most respondents (87.1%) were aware that improper antimicrobial disposal posed environmental risks, similar to 86% in a study from Ethiopia [18]. This knowledge does not always result in safe disposal practices due to factors such as limited practical understanding, weak regulatory enforcement, cultural habits, and low prioritization, emphasizing the need for targeted education and supportive policies. This study is also in concordance with a study carried out by [20] in several Middle Eastern countries, which indicated that there was limited awareness about antibiotics, which led to challenges such as difficulty recognizing antibiotics, obtaining them without proper prescriptions, and not finishing prescribed courses of antibiotic treatment. In this regard, pharmacies and healthcare providers also need to play a proactive role in educating patients about the correct ways to dispose of unwanted and leftover medicines, while emphasizing the potential health hazards of improper disposal [21].

The study findings on hazardous effects of unused antimicrobials are comparable to studies by [22,23]. Moreover, excess prescriptions contribute to the accumulation of unused medicines [23] and support the concept of antimicrobial stewardship [24]. Therefore, prescribers are encouraged to consider alternatives where appropriate and prioritize responsible antimicrobial prescriptions that will ultimately lead to effective use of antimicrobials and reduce the volume of unused medicines [24].

Consequently, a few respondents were either on 1–3 antimicrobials or more than 3 antimicrobials. This may imply the situation of polypharmacy could contribute to the stockpiling of leftover medicines in the case of patient non-adherence [25]. On the frequency of antimicrobial use by respondents, which raised concerns about potential misuse or overuse, leading to leftover medicines and improper disposal practices among some groups of people, such as students and private workers. These categories of individuals could serve as target groups for educational campaigns on rational use and disposal of medicines.

The primary source of antimicrobials for the respondents was hospitals and community pharmacies, indicating some level of regulated access, while informal sources were relatives and friends and medicine peddlers, which could thus marginally contribute to antimicrobial access and hence pose risks of inappropriate use [26] or likely substandard as well as counterfeit medicines entering circulation [27]. While most respondents purchased antimicrobials with a prescription, a notable proportion of them had access without a prescription from Over-The-Counter Medicine Sellers (OTCMS), which demonstrates that a substantial portion of the public may bypass medical guidance to access antimicrobials, with the attendant risk of irrational use [28]. This situation also signals a possibly weak regulatory enforcement system at play, leading to irrational antimicrobial use and accumulation of leftover ones, contributing to AMR [29]. Educational campaigns should address the occurrence of informal sharing of medicines among friends and family. There is a need to increase regulatory oversight on informal access to antimicrobials to prevent the circulation of substandard and/or counterfeit ones.

A lot of the respondents had leftover antimicrobials at home, which may be problematic because the conditions might not have been fully treated, ultimately increasing the risk of AMR development [30]. This calls for better patient counseling about potential side effects and how to manage them to encourage adherence. Others indicated that too many of the antimicrobials were prescribed. This highlights the need for better communication between physicians, pharmacists, and patients to ensure rational prescription and/or dispensing of antimicrobials. Providing education to both healthcare professionals and the public on the correct usage of antibiotics is an effective method to encourage responsible practices and enhance understanding of antibiotics [20].

Sharing antibiotics can promote self-medication and irrational use [30]. Many participants had not received counseling from healthcare professionals on proper antimicrobial disposal, highlighting the critical role these professionals play in guiding safe practices. Similar trends have been reported in other studies [31,32], with household trash being the most common disposal method. In Ghana, it is estimated that approximately 75% of households disposed of pharmaceutical waste through regular domestic waste bins. In contrast, findings from a recent study in Kenya indicated that the predominant disposal method for solid and semi-solid dosage forms, as well as liquid preparations, was flushing them into the water closet [33]. Both practices present considerable environmental challenges: disposal through domestic waste streams contributes to the contamination of landfills and surrounding soil, whereas flushing medicines facilitate the direct entry of pharmaceutical residues into aquatic environments, with potential ecotoxicological consequences and implications for pharmaceutical pollution. This indicates a gap between awareness and practice, driven by limited facilities, insufficient knowledge, convenience, and weak regulatory oversight. The findings underscore the need for active involvement of healthcare professionals, targeted education, supportive policies, and accessible disposal systems to ensure safe disposal and reinforce antimicrobial stewardship, as improper disposal poses risks to both the environment and public health [21,31,32].

Although many respondents had positive attitudes towards the safe use and disposal of antimicrobials, some challenges were also observed. The respondents’ lack of clarity or accessibility to adequate safe disposal information suggests that manufacturers must be able to provide more explicit guidance in the packaging inserts or patient information leaflets to improve upon users’ practices. Medicine take-back programs and community pharmacy collection schemes are internationally recognized as the most appropriate routes for the safe disposal of unused and expired medicines. Establishing such systems in Ghana would require strong collaboration between community pharmacies, hospitals, and regulatory authorities to ensure accessibility and sustainability. Integrating these programs into existing healthcare structures could not only reduce environmental contamination but also enhance public awareness and encourage responsible disposal practices [34,35,36,37].

This study further demonstrated that educational level, gender, and employment status significantly influenced respondents’ knowledge and practices regarding the disposal of medicines. Individuals with higher educational attainment and more stable finances may be more inclined to actively seek information, demonstrate greater health awareness, and possess a better understanding of their environment [38]. These factors could explain the observed variations in disposal practices across different demographic groups.

### Limitations

This study could be subject to recall bias because the respondents were asked to recall some past events; however, respondents were asked to be very honest in responding to the questions. The sample size is not representative of Ghana as a whole, and therefore the findings cannot be generalized to other populations. Furthermore, potential non-response bias (310 of the 399 respondents), reliance on self-reported practices, and restriction of this study to the Ho Municipality may affect representativeness and further limit the generalizability of the results.

## 5. Conclusions

This study revealed high awareness of the risks associated with unused and expired antimicrobials yet highlighted persistent gaps in professional advice and safe disposal practices. Demographic factors such as gender, education, and employment status influenced knowledge and behaviors, underscoring the need for targeted interventions. Strengthening public health education, implementing accessible medicine-return programs, and enforcing stricter regulations on antimicrobial sales are critical to reducing improper disposal and mitigating risks to human health and the environment.

## Figures and Tables

**Figure 1 ijerph-22-01519-f001:**
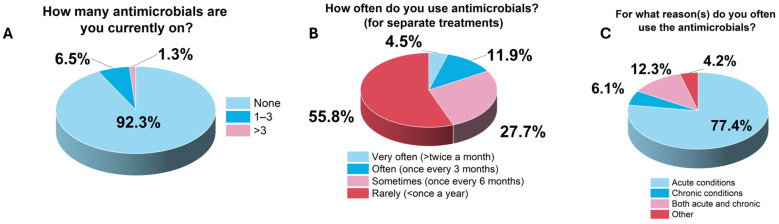
Use of antimicrobial medicines among respondents.

**Figure 2 ijerph-22-01519-f002:**
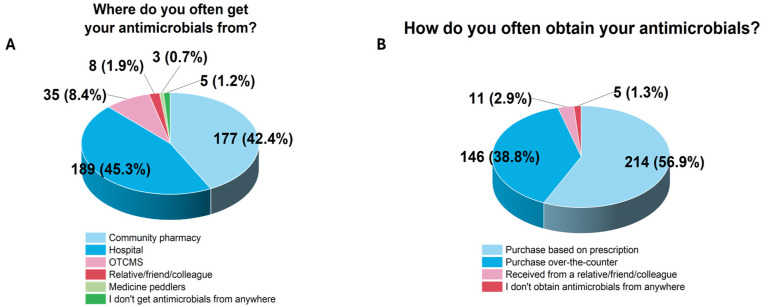
Sources of antimicrobials and how they are obtained.

**Figure 3 ijerph-22-01519-f003:**
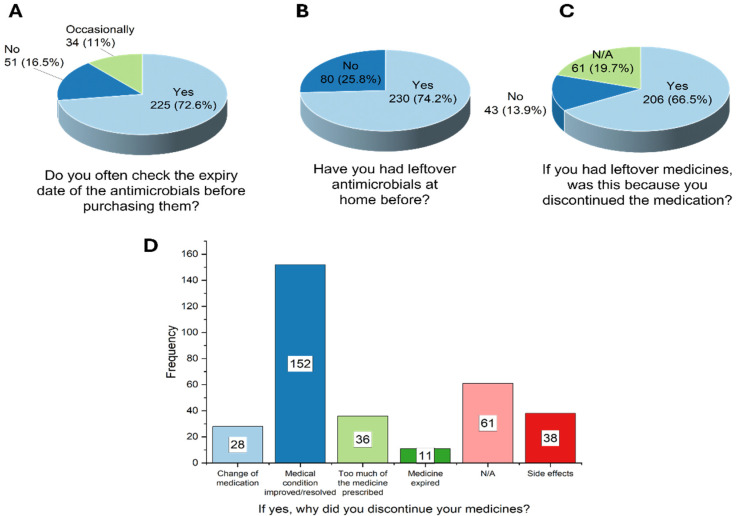
Practices in relation to antimicrobial use.

**Table 1 ijerph-22-01519-t001:** Frequency distribution of participant demographics.

Gender	Frequency	Percent (%)
Male	175	56.5
Female	135	43.5
Total	310	100.0
Age group		
18–30	196	63.2
31–40	72	23.2
41–50	30	9.7
51–60	7	2.3
>60	5	1.6
Total	310	100.0
Level of Education		
None	4	1.3
Basic/primary	18	5.8
Secondary	59	19.0
Tertiary	199	64.2
Postgraduate	30	9.7
Total	310	100.0
Employment status		
Not employed	26	8.4
Student	99	31.9
Private	74	23.9
Public	95	30.6
Self-employed	16	5.2
Total	310	100.0

**Table 2 ijerph-22-01519-t002:** Knowledge of unused medicines and associated factors.

Knowledge Statement	Agree N (%)	Disagree N (%)	Not Sure N (%)	Total N (%)	Logistic Regression		
					Significant Demographic Predictors	Adjusted Odds Ratio (95% CI)	*p*-Value
A. Failing to complete antibiotics can contribute to drug resistance	268 (86.5%)	23 (7.4%)	19 (6.1%)	310 (100%)	None significant	-	-
B. Improper disposal of unused antimicrobials can affect the environment	270 (87.1%)	17 (5.5%)	23 (7.4%)	310 (100%)	Private employment (borderline)	0.56 (0.31–1.01)	0.052
C. Improper disposal of unused antimicrobials can affect health	259 (83.5%)	24 (7.7%)	27 (8.7%)	310 (100%)	Female sex Higher education Self-employed	1.53 (1.01–2.32) 1.22 (1.02–1.45) 0.51 (0.26–0.99)	0.046 0.027 0.049
D. Antimicrobial residues in water bodies can affect aquatic life	283 (91.3%)	5 (1.6%)	22 (7.1%)	310 (100%)	None significant	-	-
E. Unused or leftover antimicrobials present potential risks at home	266 (85.8%)	17 (5.5%)	27 (8.7%)	310 (100%)	Private employment Self-employed	0.54 (0.30–0.96) 0.52 (0.27–0.99)	0.034 0.049
F. Children are the most vulnerable to risks associated with unused antimicrobials	283 (91.3%)	13 (4.2%)	14 (4.5%)	310 (100%)	Female sex Higher education	1.62 (1.07–2.45) 1.20 (1.00–1.43)	0.023 0.049

**Table 3 ijerph-22-01519-t003:** Management of the hazardous effects of unused antimicrobials and their associated factors.

The Hazardous Effects of Unused Antimicrobials Can Be Minimized or Controlled By	Responses		Logistic Regression		
	N	Percent	Significant Demographic Predictors	Adjusted Odds Ratio (95% CI)	*p*-Value
Providing proper guidance to the consumer	283	91.30%	None significant	-	-
Prescribing adequate quantities for patient compliance	218	70.3%	Higher education	1.29 (1.03–1.62)	0.027
Reducing the number of prescribed antimicrobials by the doctor	138	44.5%	Higher education Private employment Student status Self-employed	1.47 (1.16–1.84) 0.41(0.19–0.88) 0.46 (0.23–0.92) 0.32 (0.14–0.78)	0.001 0.022 0.026 0.012
Donating or sharing the unused antimicrobials	27	8.7%	Higher education	0.75 (0.57–0.99)	0.045
Total (N)	310				

**Table 4 ijerph-22-01519-t004:** Disposal practices of antimicrobial drugs.

Practice	Frequency	Percent *	Logistic Regression		
			Significant Demographic Predictors	Adjusted Odds Ratio (95% CI)	*p*-Value
How often do you share leftover antimicrobials with people with similar symptoms as yours?					
Always	13	4.2%	Higher education	0.62 (0.40–0.98)	0.040
Sometimes	87	28.1%			
Never	210	67.7%			
Total (N)	310	100.0			
Have you ever disposed of any antimicrobial drug?					
Yes	234	75.5%	None significant	-	-
No	76	24.5%			
Total (N)	310	100.0			
Have you ever been advised by a healthcare professional about proper antimicrobial disposal?					
Yes	77	24.8%	Higher education	1.35 (1.06–1.70)	0.014
No	223	71.9%			
Not sure	10	3.2%			
Total (N)	310	100.0			
In what forms do the leftover antimicrobials usually appear? ^α^					
Tablets/capsules	265	85.5%			
Ointments/creams	64	20.6%			
Suppositories/vaginal pessaries	26	8.4%			
Syrups/suspensions	49	15.8%			
N/A	30	9.7%			
Total (N)	310				
If yes, what method(s) did you use to dispose of the following dosage forms of antimicrobials? Tablets/Capsules ^α^					
Burn	55	17.7%	Older age Higher education	1.32 (1.04–1.68) 0.73 (0.58–0.92)	0.023 0.008
Return to a health facility (hospital/pharmacy)	7	2.3%			
Flush down the toilet/sink	16	5.2%			
Household trash	202	65.2%	Higher education Self-employed	0.82 (0.68–0.99) 1.89 (1.05–3.41)	0.038 0.034
Others: I have not	3	1.0%			
Total (N)	310				
Syrups/Suspensions ^α^					
Burn	13	4.2%			
Household trash	109	35.2%	Female sex	1.62 (1.02–2.57)	0.041
Flush down the toilet/sink	35	11.3%			
Return to health facility (hospital/pharmacy)	7	2.3%	Higher education	1.85 (1.12–3.06)	0.016
Others: pour away and put the bottle in the trash	4	1.3%			
Total (N)	310				
Ointments/Creams ^α^					
Household trash	132	42.6%	Privately employed	0.52 (0.28–0.97)	0.039
Burn	21	6.8%			
Flush down the toilet/sink	9	2.9%			
Return to health facility (hospital/pharmacy)	9	2.9%	Higher education	2.14 (1.18–3.88)	0.012
Others: FDA	4	1.3%			
Total (N)	310				
Suppositories/Vaginal Pessaries ^α^					
Household trash	105	33.9%	Older age	0.76 (0.59–0.98)	0.035
Burn	11	3.5%			
Return to health facility (hospital/pharmacy)	8	2.6%	Higher education	2.37 (1.29–4.35)	0.005
Flush down the toilet/sink	10	3.2%			
Others: FDA	2	0.6%			
Total (N)	310				

* Percentages based on total respondents (N = 310). **^α^** Multiple responses possible; percentages represent proportion of total respondents selecting each option. Reference categories: male sex, lower education, not employed.

**Table 5 ijerph-22-01519-t005:** Attitude and perception on unused or leftover medicine disposal.

Attitude and Perception on Unused or Leftover Medicine Disposal	Agree	Disagree		Not Sure	
	N (%)	N (%)		N (%)	
Purchasing antimicrobials without a prescription can add to the number of unused medicines at home.	273 (88.1%)	21 (6.8%)		16 (5.2%)	
Is there adequate information on the safe disposal of unused antimicrobials from medicine manufacturers (inserts)?	149 (48.1%)	119 (38.4%)		42 (13.5%)	
Who do you think is/are responsible for creating awareness for the proper disposal of unused antimicrobials? *			Responses		
			N		Percent
Government agencies			162		52.3%
Pharmaceutical industries			219		70.6%
Public			100		32.3%
Pharmacist			223		71.9%
Other healthcare professionals			183		59.0%
None of the above			5		1.6%
Total			310		
Are you aware of institutions that collect unused medicines?			N		Percent
Yes			76		24.5
No			234		75.5
Total			310		100.0
If yes, which institutions do you know of? *			N		Percent
Government agencies (e.g., FDA)			51		67.1%
Hospitals			38		50.0%
Pharmacies			43		56.6%
Others (Environmental Protection Agency, NGOs)			10		13.2%
Total			76		
Have you ever participated in any medicine-return program?			Frequency		Percent
Yes			25		8.1
No			228		73.5
Not Aware			57		18.4
Total			310		100.0
Would you be willing to participate in a medicine-return program if it is available near you?					
Yes			284		91.6%
No			26		8.4%
Total			310		100.0
Take-back programs for unused/leftover medicines should be made mandatory.					
Agree			199		64.2%
Disagree			78		25.2%
Not sure			33		10.6%
Total			310		100.0
Where would you prefer to return your unused/leftover antimicrobials? *			N		Percent
Community pharmacy			160		51.6%
Hospital/CHPS facility			154		49.7%
Home pickup			105		33.9%
Shopping mall/supermarket			21		6.8%
None of the above			27		8.7%
Total			310		

* Respondents were asked to choose more than one option if applicable.

## Data Availability

The data that were used and analyzed to support the findings of this study are available from the corresponding author upon reasonable request.

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
