# Peer review of "Knowledge and Practices on Household Disposal of Unused Antimicrobials in Ho Municipality, Ghana"

_ijerph, 2025, doi:10.3390/ijerph22101519_

Round 1
Reviewer 1 Report
Comments and Suggestions for Authors
The manuscript titled “Public Knowledge, Awareness and Practices on Domestic Disposal of Unused and Expired Antimicrobials” presents a cross-sectional survey conducted in the Ho Municipality, Ghana, assessing the public knowledge, awareness, and practices regarding the disposal of unused and expired antimicrobials. The study addresses a highly relevant public health issue within the broader context of antimicrobial resistance (AMR), contributing useful data from a peri-urban African setting, which is often underrepresented in the literature.
The abstract is clear, informative, and well-structured. It summarizes the rationale, objectives, methods, key results, and conclusions effectively.
“This study was carried out to assess the knowledge, awareness, and practices of residents…”
This is a generic formulation that does not clearly convey the rationale or the research gap.
Specify what gap this study addresses (e.g., lack of data in peri-urban Ghana), and why KAP (knowledge-attitude-practice) assessment is a useful method.
The introduction is comprehensive and well-referenced. It provides a solid contextual foundation by linking inappropriate disposal of antimicrobials to environmental risks and the global threat of AMR. The authors appropriately highlight the lack of research on medicine disposal practices in peri-urban Ghana, justifying the study’s relevance.
However, the introduction relies heavily on widely cited global data without critically analyzing or contextualizing them:
“The improper disposal of unused and expired medications… poses significant public health and environmental risks (1).”
“These practices also exacerbate the global antimicrobial resistance (AMR) crisis (6,7).”
While accurate, these statements are generic and reflect a literature summary more than a critical synthesis. Provide a more targeted discussion that contrasts findings from urban vs. rural or developed vs. developing countries and emphasize why the Ghanaian peri-urban context deserves attention.
The Materials and Methods section is generally well described. The study design, inclusion criteria, sampling method (Slovin’s formula), and data collection approach are adequately detailed. However, while the calculated sample size was 399, only 310 participants were ultimately included, with no explanation provided for this discrepancy. This should be addressed to clarify potential selection bias or recruitment limitations.
The Results section is clearly organized, with relevant tables and figures that enhance understanding. The descriptive statistics are well presented, and key patterns in disposal behavior and public awareness are effectively highlighted. The use of cross-tabulation and chi-square tests adds some analytical depth, though more robust statistical analysis could strengthen the findings.
Concerning the discussion section, while the authors reference numerous studies from other countries, the comparisons are shallow and non-analytical.
Phrases like:
“These findings are consistent with a study in Ethiopia…”
“A study in Indonesia also found…”
do little to explain why findings are similar or different. There’s no examination of contextual factors—e.g., regulatory systems, healthcare infrastructure, cultural norms. Synthesize findings from the literature, highlight contrasts, and critically explore what they imply for Ghana’s public health policies.
The discussion of limitations is present but could be slightly expanded. The authors mention recall bias and the limited generalizability of findings, which is accurate. However, a brief note on the cross-sectional nature of the study and its inability to establish causality would further strengthen this section.
The conclusions are consistent with the data and appropriately highlight the need for structured disposal programs, increased public awareness, and regulatory action to control antimicrobial misuse and environmental contamination. The implications are relevant and practical.
Author Response
Comment 1: “This study was carried out to assess the knowledge, awareness, and practices of residents…” This is a generic formulation that does not clearly convey the rationale or the research gap. Specify what gap this study addresses (e.g., lack of data in peri-urban Ghana), and why KAP (knowledge-attitude-practice) assessment is a useful method. |
Response 1: Thank you for your comments. The statement has been revised. Kindly refer to the Abstract on Page 2 and Lines 23-25. |
Comment 2: The introduction is comprehensive and well-referenced. It provides a solid contextual foundation by linking inappropriate disposal of antimicrobials to environmental risks and the global threat of AMR. The authors appropriately highlight the lack of research on medicine disposal practices in peri-urban Ghana, justifying the study’s relevance. However, the introduction relies heavily on widely cited global data without critically analyzing or contextualizing them: “The improper disposal of unused and expired medications… poses significant public health and environmental risks (1).” “These practices also exacerbate the global antimicrobial resistance (AMR) crisis (6,7).” While accurate, these statements are generic and reflect a literature summary more than a critical synthesis. Provide a more targeted discussion that contrasts findings from urban vs. rural or developed vs. developing countries and emphasize why the Ghanaian peri-urban context deserves attention. |
Response 2: Agree. We have, accordingly, modified the statements to emphasize this point. Refer to Pages 3 and 4 and Lines 48 to 94. Comment 3: The Materials and Methods section is generally well described. The study design, inclusion criteria, sampling method (Slovin’s formula), and data collection approach are adequately detailed. However, while the calculated sample size was 399, only 310 participants were ultimately included, with no explanation provided for this discrepancy. This should be addressed to clarify potential selection bias or recruitment limitations. Response 3: The statements on the sample size has been updated. Refer to Page 6 and Lines 120 to 123. |
Comment 4: The Results section is clearly organized, with relevant tables and figures that enhance understanding. The descriptive statistics are well presented, and key patterns in disposal behavior and public awareness are effectively highlighted. The use of cross-tabulation and chi-square tests adds some analytical depth, though more robust statistical analysis could strengthen the findings.
Response 4: Logistic regression has been done. Refer to the results section from Pages 9 to 18.
Comment 5: Concerning the discussion section, while the authors reference numerous studies from other countries, the comparisons are shallow and non-analytical.
Phrases like:
“These findings are consistent with a study in Ethiopia…”
“A study in Indonesia also found…”
do little to explain why findings are similar or different. There’s no examination of contextual factors—e.g., regulatory systems, healthcare infrastructure, cultural norms. Synthesize findings from the literature, highlight contrasts, and critically explore what they imply for Ghana’s public health policies.
Response 5: The discussion has revised. Refer to Pages 21 to 25.
Reviewer 2 Report
Comments and Suggestions for Authors
In the submitted manuscript, the authors report a cross-sectional study about "the knowledge, attitudes, and practices of households on how they dispose of unused and leftover antimicrobials.
The study is valid for the education and normalization of practices regarding the safe disposal of medicines. The study examines the behavior of the local population in this matter.
Recommendations for this article are to include the class of antibiotic, antimicrobials consumed for the investigated population, the estimated quantity of antimicrobial consumed and discarded in the local community, for the sake of illustrating the flow of substance (antimicrobials) in the environment.
Additionally, it is recommended that the authors exemplify how antimicrobials in the environment are detrimental to all living beings, including humans, domestic animals, microbiomes, and wildlife.
Author Response
In the submitted manuscript, the authors report a cross-sectional study about "the knowledge, attitudes, and practices of households on how they dispose of unused and leftover antimicrobials. The study is valid for the education and normalization of practices regarding the safe disposal of medicines. The study examines the behavior of the local population in this matter. Comment 1: Recommendations for this article are to include the class of antibiotic, antimicrobials consumed for the investigated population, the estimated quantity of antimicrobial consumed and discarded in the local community, for the sake of illustrating the flow of substance (antimicrobials) in the environment. |
Response 1: Thank you for this insightful suggestion. We agree that including information on the specific classes of antibiotics, the types of antimicrobials consumed, and the estimated quantities consumed and discarded would provide a more comprehensive understanding of antimicrobial flow within the community and its potential environmental implications. Unfortunately, such data were not captured in the present study. |
Comment 2: Additionally, it is recommended that the authors exemplify how antimicrobials in the environment are detrimental to all living beings, including humans, domestic animals, microbiomes, and wildlife. |
Response 2: Agree. We have, accordingly, modified emphasizing the above recommendation in the introduction. Kindly refer to Introduction Page 2 Lines 48 to 56. |
Reviewer 3 Report
Comments and Suggestions for Authors
I evaluated the PDF article titled “Public Knowledge, Awareness and Practices on Domestic Disposal of Unused and Expired Antimicrobials”. Below are revision points that would strengthen the manuscript’s academic rigour, clarity and relevance.
Title and Abstract
Clarify the focus – the current title mixes “public knowledge” with “domestic disposal” and does not specify the target population or location. A clearer title could be “Knowledge and Practices on Household Disposal of Unused Antimicrobials in Ho Municipality, Ghana”.
Refine the abstract – avoid including minor procedural details (e.g., software names) and emphasise the key findings and implications. The abstract could end with a clear recommendation (e.g., “most respondents dispose of antimicrobials in household trash; take‑back programs and community‑pharmacy collection are needed”).
Introduction
Better link the research gap – the introduction should clearly explain why investigating disposal practices in peri‑urban Ho is necessary. Current text lists global AMR issues but barely addresses local waste‑management gaps. Strengthen this by referring to evidence that improper disposal can be a significant environmental pathway; unused medicines flushed down toilets or sent to landfill can contaminate water bodies.
Use up‑to‑date guidelines – to support your rationale, cite public health recommendations. For example, the U.S. FDA states that the best way to discard unused medicines is through drug take‑back programs, with mailing options when drop‑off sites are unavailable.
Methods
Sample size calculation – the paper states that Slovin’s formula was used, but it does not provide the margin of error (e) or justification. Clarify these values, and discuss limitations. Slovin’s formula is a simple heuristic; it assumes a homogeneous population and is appropriate only for simple random sampling. If the population has subgroups (e.g., age or education strata), Slovin’s formula may produce biased estimates. Consider using standard prevalence‑based sample size formulas or cluster/stratified sampling techniques for better precision.
Sampling and recruitment – describe how households were selected (random, convenience or multistage). The method currently implies convenience sampling, which limits representativeness. If random sampling was not possible, explain why and acknowledge potential bias.
Questionnaire validation – you mention adapting questions from previous studies, translating them into local languages and pretesting with 20 participants. Include information on reliability testing (e.g., Cronbach’s alpha) and whether the translations were back‑translated to ensure accuracy.
Ethics – confirm that informed consent was obtained and that anonymity was maintained; complete the unfinished sentence (“Electronic data were stored on password‑…”) so that readers understand data protection measures.
Results
Improve tables and figures – some tables are difficult to interpret due to long captions and inconsistent formatting. Ensure table titles are concise and footnotes explain abbreviations. Provide figure legends that describe what each chart depicts (e.g., “Figure 1. Number of antimicrobials currently used by respondents”).
Report statistical tests correctly – you use χ² tests, but some sentences omit test statistics or p‑values. Provide these in brackets (e.g., “χ²=485.6, p<0.0001”) consistently. When comparing knowledge or practice across demographics, consider logistic regression to control for confounders instead of only descriptive statistics.
Clarify numbers and percentages – ensure that percentages add up logically. In Table 4, for instance, the proportions of responses should be consistent with the number of respondents who answered each question.
Discussion
Interpret findings more critically – the discussion mostly reiterates the results and briefly compares them to previous studies. Highlight why certain disposal methods (e.g., throwing antimicrobials into household trash) are problematic by linking back to environmental literature: unused pharmaceuticals disposed of via sinks or household waste can become a significant route of contamination.
Explain the impact of knowledge gaps – many respondents recognised the environmental risk of improper disposal, yet a high proportion had never been counselled on disposal. Discuss why knowledge does not always translate into correct behaviour and how health professionals could bridge this gap.
Relate findings to policy and practice – emphasise that drug take‑back programs or community‑pharmacy collection schemes are the recommended disposal route. Suggest collaboration between pharmacies, hospitals and regulators to establish accessible return programs in Ghana.
Limitations
Expand the limitations section – acknowledge not only recall bias and sample size but also potential non‑response bias (only 310 of the targeted 399 participated), use of self‑reported practices, and limited generalisability beyond Ho Municipality.
References
Ensure accuracy and formatting – check that all citations in the text correspond to the numbered references and follow the journal’s reference style. For instance, italicise journal titles and give complete page ranges.
Include up‑to‑date sources – add more recent literature on pharmaceutical waste management and policies to strengthen the discussion. For example, the FDA’s guidelines on medicine disposal and environmental research show the role of disposal in contamination.
Author Response
I evaluated the PDF article titled “Public Knowledge, Awareness and Practices on Domestic Disposal of Unused and Expired Antimicrobials”. Below are revision points that would strengthen the manuscript’s academic rigour, clarity and relevance. Comment 1: Title and Abstract Clarify the focus – the current title mixes “public knowledge” with “domestic disposal” and does not specify the target population or location. A clearer title could be “Knowledge and Practices on Household Disposal of Unused Antimicrobials in Ho Municipality, Ghana”. |
Response 1: Thank you for pointing this out. We agree with this comment. Therefore, we have modified the title. Please refer to Page 1. |
Comment 2: Refine the abstract – avoid including minor procedural details (e.g., software names) and emphasise the key findings and implications. The abstract could end with a clear recommendation (e.g., “most respondents dispose of antimicrobials in household trash; take back programs and community pharmacy collection are needed”). |
Response 2: Agree. We have, accordingly, revised the abstract. Refer to Page 2 to 3. Lines 29 to 30. |
Comment 3: Introduction
Better link the research gap – the introduction should clearly explain why investigating disposal practices in peri urban Ho is necessary. Current text lists global AMR issues but barely addresses local waste management gaps. Strengthen this by referring to evidence that improper disposal can be a significant environmental pathway; unused medicines flushed down toilets or sent to landfill can contaminate water bodies.
Use up to date guidelines – to support your rationale, cite public health recommendations. For example, the U.S. FDA states that the best way to discard unused medicines is through drug take back programs, with mailing options when drop off sites are unavailable.
Response 3: The introduction has been revised. Refer to Page 3 to 5
Comment 4: Methods
- Sample size calculation – the paper states that Slovin’s formula was used, but it does not provide the margin of error (e) or justification. Clarify these values, and discuss limitations. Slovin’s formula is a simple heuristic; it assumes a homogeneous population and is appropriate only for simple random sampling. If the population has subgroups (e.g., age or education strata), Slovin’s formula may produce biased estimates. Consider using standard prevalence based sample size formulas or cluster/stratified sampling techniques for better precision.
Response 4a: We appreciate this comment. In the revised manuscript, we have specified that Slovin’s formula was applied using a 5% margin of error (e = 0.05). The limitations have been addressed in the Limitations sections. Although prevalence-based sample size formulas or stratified/cluster sampling methods could have provided more precise estimates, logistical and resource constraints restricted our use of these approaches. Nevertheless, we believe that the calculated sample size was sufficient to generate valuable insights into antimicrobial disposal practices in the study population.
- Sampling and recruitment – describe how households were selected (random, convenience or multistage). The method currently implies convenience sampling, which limits representativeness. If random sampling was not possible, explain why and acknowledge potential bias.
Response 4b: We are grateful for this observation however we have included in the revised manuscript that households were selected using a convenience sampling. This approach was due to limited resources and logistical constraints, which made random or multistage sampling infeasible. We acknowledge that this method may limit representativeness and introduce potential selection bias which have been stated under limitations .To reduce this risk, households from varied neighborhoods and socio-demographic groups were included. Refer to Page 6 Line 130
- Questionnaire validation – you mention adapting questions from previous studies, translating them into local languages and pretesting with 20 participants. Include information on reliability testing (e.g., Cronbach’s alpha) and whether the translations were back translated to ensure accuracy.
Response 4c: Thank you for your comment. The questionnaire was adapted from previously validated tools, translated into local languages, and then back-translated into English by bilingual experts to ensure accuracy. Refer to Page 7 Lines 140 to 142.
- Ethics – confirm that informed consent was obtained and that anonymity was maintained; complete the unfinished sentence (“Electronic data were stored on password …”) so that readers understand data protection measures.
Response 4d: This statement has been updated. Refer to Page 7 Lines 147-148 and 155-157
Comment 5 : Results
- Improve tables and figures – some tables are difficult to interpret due to long captions and inconsistent formatting. Ensure table titles are concise and footnotes explain abbreviations. Provide figure legends that describe what each chart depicts (e.g., “Figure 1. Number of antimicrobials currently used by respondents”).
Response 5a: Thank you, reviewer for this observation. This information has already been addressed in the manuscript. Refer to Page 13.
- Report statistical tests correctly – you use χ² tests, but some sentences omit test statistics or p values. Provide these in brackets (e.g., “χ²=485.6, p<0.0001”) consistently. When comparing knowledge or practice across demographics, consider logistic regression to control for confounders instead of only descriptive statistics.
Response 5b: This comment has been addressed. Refer to Results section. Page 8.
- Clarify numbers and percentages – ensure that percentages add up logically. In Table 4, for instance, the proportions of responses should be consistent with the number of respondents who answered each question.
Response 5c: Table 4 has been updated. Refer to Pages 17-18.
Comment 6: Discussion
- Interpret findings more critically – the discussion mostly reiterates the results and briefly compares them to previous studies. Highlight why certain disposal methods (e.g., throwing antimicrobials into household trash) are problematic by linking back to environmental literature: unused pharmaceuticals disposed of via sinks or household waste can become a significant route of contamination.
Response 6a: The discussion has been revised accordingly. Refer to Page 21-25
- Explain the impact of knowledge gaps – many respondents recognised the environmental risk of improper disposal, yet a high proportion had never been counselled on disposal. Discuss why knowledge does not always translate into correct behaviour and how health professionals could bridge this gap.
Response 6b: The discussion has been modified to include the above recommendations. Kindly refer to Page 21 to
- Relate findings to policy and practice – emphasise that drug take back programs or community pharmacy collection schemes are the recommended disposal route. Suggest collaboration between pharmacies, hospitals and regulators to establish accessible return programs in Ghana.
Response 6c: The section on the discussion has been updated. Refer to Pages 21 to 25.
Comment 7: Limitations
Expand the limitations section – acknowledge not only recall bias and sample size but also potential non response bias (only 310 of the targeted 399 participated), use of self reported practices, and limited generalisability beyond Ho Municipality.
Response 7: The limitations have been revised. Refer to Page 25
Comment 8 :References
Ensure accuracy and formatting – check that all citations in the text correspond to the numbered references and follow the journal’s reference style. For instance, italicise journal titles and give complete page ranges.
Include up to date sources – add more recent literature on pharmaceutical waste management and policies to strengthen the discussion. For example, the FDA’s guidelines on medicine disposal and environmental research show the role of disposal in contamination.
Response 8: More references have been included. Refer to Discussion from Pages and References.
Round 2
Reviewer 2 Report
Comments and Suggestions for Authors
The study is valid for the education and normalization of practices regarding the safe disposal of medicines. The study examines the behavior of the local population regarding their knowledge, attitudes, and practices regarding the disposal of unused and leftover antimicrobials.
In the revised version, although the authors responded only to one of the critical points addressed by the reviewer, justified by the conceptual study approach and the finalization of data collection, the article was improved and is now acceptable for publication.